# WHEN, WHY, AND WHICH PRETRAINED GANS ARE USEFUL?

**Timofey Grigoryev** [*]
Yandex
`grigorev.ta@phystech.edu`

**Andrey Voynov** [*]
Yandex
`an.voynov@yandex.ru`

**Artem Babenko**
Yandex
`artem.babenko@phystech.edu`

## ABSTRACT

The literature has proposed several methods to finetune pretrained GANs on new datasets, which typically results in higher performance compared to training from scratch, especially in the limited-data regime. However, despite the apparent empirical benefits of GAN pretraining, its inner mechanisms were not analyzed in-depth, and understanding of its role is not entirely clear. Moreover, the essential practical details, e.g., selecting a proper pretrained GAN checkpoint, currently do not have rigorous grounding and are typically determined by trial and error.

This work aims to dissect the process of GAN finetuning. First, we show that initializing the GAN training process by a pretrained checkpoint primarily affects the model's coverage rather than the fidelity of individual samples. Second, we explicitly describe how pretrained generators and discriminators contribute to the finetuning process and explain the previous evidence on the importance of pretraining both of them. Finally, as an immediate practical benefit of our analysis, we describe a simple recipe to choose an appropriate GAN checkpoint that is the most suitable for finetuning to a particular target task. Importantly, for most of the target tasks, Imagenet-pretrained GAN, despite having poor visual quality, appears to be an excellent starting point for finetuning, resembling the typical pretraining scenario of discriminative computer vision models.

## 1 INTRODUCTION

These days, generative adversarial networks (GANs) (Goodfellow et al., 2014) can successfully approximate the high-dimensional distributions of real images. The exceptional quality of the state-of-the-art GANs (Karras et al., 2020b; Brock et al., 2019) makes them a key ingredient in applications, including semantic editing (Isola et al., 2017; Zhu et al., 2018; Shen et al., 2020; Voynov & Babenko, 2020), image processing (Pan et al., 2020; Ledig et al., 2017; Menon et al., 2020), video generation (Wang et al., 2018a), producing high-quality synthetics (Zhang et al., 2021; Voynov et al., 2020).

To extend the success of GANs to the limited-data regime, it is common to use pretraining, i.e., to initialize the optimization process by the GAN checkpoint pretrained on some large dataset. A line of works (Wang et al., 2018b; Noguchi & Harada, 2019; Zhao et al., 2020; Mo et al., 2020; Wang et al., 2020; Li et al., 2020) investigate different methods to transfer GANs to new datasets and report significant advantages compared to training from scratch both in terms of generative quality and convergence speed. However, the empirical success of GAN pretraining was not investigated in-depth, and its reasons are not entirely understood. From the practical standpoint, it is unclear how to choose a proper pretrained checkpoint or if one should initialize both generator and discriminator or only one of them. To the best of our knowledge, the only work that systematically studies the benefits of pretraining is Wang et al. (2018b). However, the experiments in (Wang et al., 2018b) were performed with currently outdated models, and we observed that some conclusions from Wang

---

[*]Indicates equal contribution

et al. (2018b) are not confirmed for modern architectures like StyleGAN2 (Karras et al., 2020b). In particular, unlike the prior results, it appears that for state-of-the-art GANs, it is beneficial to transfer from sparse and diverse sources rather than dense and less diverse ones.

In this work, we thoroughly investigate the process of GAN finetuning. First, we demonstrate that starting the GAN training from the pretrained checkpoint can significantly influence the diversity of the finetuned model, while the fidelity of individual samples is less affected. Second, we dissect the mechanisms of how pretrained generators and discriminators contribute to the higher coverage of finetuned GANs. In a nutshell, we show that a proper pretrained generator produces samples in the neighborhood of many modes of the target distribution, while a proper pretrained discriminator serves as a gradient field that guides the samples to the closest mode, which together result in a smaller risk of mode missing. This result explains the evidence from the literature that it is beneficial to initialize both generator and discriminator when finetuning GANs. Finally, we investigate different ways to choose a suitable pretrained GAN checkpoint for a given target dataset. Interestingly, for most of the tasks, Imagenet-pretrained models appear to be the optimal initializers, which mirrors the pretraining of discriminative models, where Imagenet-based initialization is de-facto standard (Donahue et al., 2014; Long et al., 2015; He et al., 2020; Chen et al., 2020a). Our conclusions are confirmed by experiments with the state-of-the-art StyleGAN2 (Karras et al., 2020b), chosen due to its practical importance and a variety of open-sourced checkpoints, which can be used as pretrained sources. The code and pretrained models are available online at https://github.com/yandex-research/gan-transfer

The main contributions of our analysis are the following:

1. We show that initializing the GAN training by the pretrained checkpoint can significantly affect the coverage and has much less influence on the realism of individual samples.

2. We explain why it is important to initialize both generator and discriminator by describing their roles in the finetuning process.

3. We describe a simple automatic approach to choose a pretrained checkpoint that is the most suitable for a given generation task.

## 2   ANALYSIS

This section aims to explain the success of the GAN finetuning process compared to training from scratch. First, we formulate the understanding of this process speculatively and then confirm this understanding by experiments on synthetic data and real images.

### 2.1   HIGH-LEVEL INTUITION

Let us consider a pretrained generator $G$ and discriminator $D$ that are used to initialize the GAN training on a new data from a distribution $p_{\text{target}}$. Throughout the paper, we show that a discriminator initialization is "responsible for" an initial gradient field, and a generator initialization is "responsible for" a target data modes coverage. Figure 1 illustrates the overall idea with different initialization patterns. Intuitively, the proper discriminator initialization guarantees that generated samples will move towards "correct" data regions. On the other hand, the proper pretrained generator guarantees that the samples will be sufficiently diverse at the initialization, and once guided by this vector field, they will cover all target distribution. Below, we confirm the validity of this intuition.

### 2.2   SYNTHETIC EXPERIMENT

We start by considering the simplest synthetic data presented on Figure 2. Our goal is to train a GAN on the target distribution, a mixture of ten Gaussians arranged in a circle. We explore three options to initialize the GAN training process. First, we start from random initialization. The second and the third options initialize training by GANs pretrained on the two different source distributions. The first source distribution corresponds to a wide ring around the target points, having high coverage and low precision w.r.t. target data. The second source distribution is formed by three Gaussians that share their centers with three target ones but have a slightly higher variance. This source distribution has high precision and relatively low coverage w.r.t. target data. Then we train two source GANs from scratch to fit the first and the second source distributions and employ these checkpoints to initialize GAN training on the target data. The results of GAN training for the three options are presented on Figure 2, which shows the advantage of pretraining from the more diverse model,

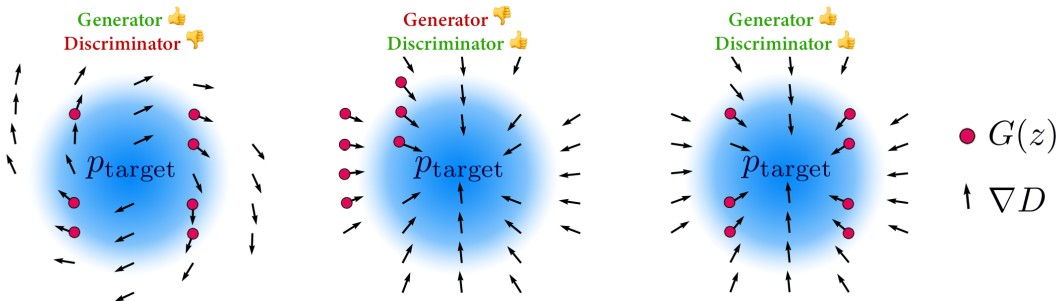

Figure 1: Different $G/D$ initialization patterns: the red dots denote pretrained generator samples, the arrows denote a pretrained discriminator gradient field, the blue distribution is the target. From left to right: bad discriminators will lead good initial samples out of the target distribution; bad generators will drop some of the modes even being guided by good discriminators; both proper $G/D$ serve as an optimal initialization for transfer to a new task.

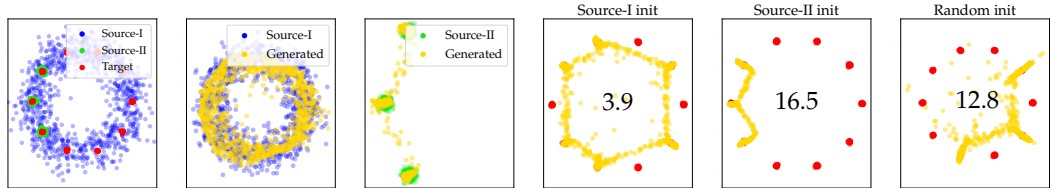

Figure 2: Impact of GAN pretraining for synthetic data. *1*) source and target distributions. *2-3*) GANs pretrained on two source distributions. *4-6*): GANs trained on the target distribution, initialized by the two source checkpoints and randomly. Each plot also reports the Wasserstein-1 distance between the generated and the target distributions.

which results in a higher number of covered modes. The details of the generation of the synthetic are provided in the appendix.

**Dissecting the contributions from $G$ and $D$.** Here, we continue with the synthetic example from above and take a closer look at the roles that the pretrained generator and discriminator play when finetuning GANs. Our goal is to highlight the importance of (1) the initial coverage of the target distribution by the pretrained generator and (2) the quality of the gradient field from the pretrained discriminator. We quantify the former by the established recall measure (Kynkäänniemi et al., 2019) computed in the two-dimensional dataspace with $k=5$ for 1000 randomly picked samples from the target distribution and the same number of samples produced by the pretrained generator. To evaluate the quality of the discriminator gradient field, we use a protocol described in (Sinha et al., 2020). Namely, we assume that the "golden" ground truth gradients would guide each sample towards the closest Gaussian center from the target distribution. Then we compute the similarity between the vector field $\nabla_x D$ provided by the pretrained discriminator and the vector field of "golden" gradients. Specifically, we evaluate the cosine similarity between these vector fields, computed for the generated samples.

Given these two measures, we consider a series of different starting generator/discriminator checkpoints $(G_i, D_i), i = 1, \ldots, N$. The details on the choice of the starting checkpoints are provided in the appendix. Then we use each pair $(G_i, D_i)$ as initialization of GAN training on the target distribution of ten Gaussians described above. Additionally, for all starting $G_i/D_i$, we evaluate the recall and the discriminator gradients field similarity to the "golden" gradients. The overall quality of GAN finetuning is measured as the Wasserstein-1 distance between the target distribution and the distribution produced by the finetuned generator. The scatter plots of recall, the similarity of gradient fields, and Wasserstein-1 distance are provided in Figure 3. As can be seen, both the recall and gradient similarity have significant negative correlations with the $W_1$-distance between the ground-truth distribution and the distribution of the finetuned GAN. Furthermore, for the same level of recall, the higher values of the gradient similarity correspond to lower Wasserstein distances. Alternatively, for the same value of gradient similarity, higher recall of the source generator typically corresponds to the lower Wasserstein distance. We also note that the role of the pretrained generator is more important since, for high recall values, the influence from the discriminator is not significant (see Figure 3, left).

This synthetic experiment does not rigorously prove the existence of a causal relationship between the recall or gradient similarity and the quality of the finetuned GANs since it demonstrates only correlations of them. However, in the experimental section, we show that these correlations can be successfully exploited to choose the optimal pretraining checkpoint, even for the state-of-the-art GAN architectures.

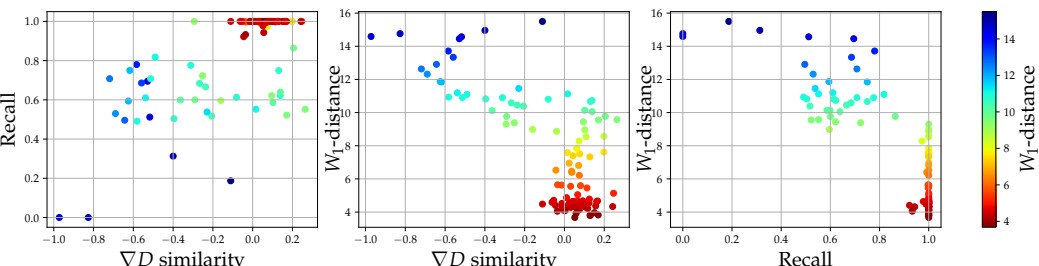

Figure 3: Scatter plots of the pretrained generator quality (*Recall*) and the pretrained discriminator quality ($\nabla D$ *similarity*) vs the quality of finetuned GAN ($W_1$-*distance*). Each point represents a result of GAN finetuning, which started from a particular pair of pretrained discriminator and generator. The color indicates the $W_1$-distance between the final generator distribution and the target distribution. The Pearson correlation of the final $W_1$-distance is equal $-0.84$ for the Recall, and $-0.73$ for the gradient similarity.

## 3 LARGE-SCALE EXPERIMENTS

### 3.1 EXPLORING PRETRAINING FOR STYLEGAN2

In this section, we confirm the conclusions from the previous sections experimentally with the state-of-the-art StyleGAN2 architecture (Karras et al., 2020b). If not stated otherwise, we always work with the image resolution $256 \times 256$.

**Datasets.** We work with six standard datasets established in the GAN literature. We also include two datasets of satellite images to investigate the pretraining behavior beyond the domain of natural images. As potential pretrained sources, we use the StyleGAN2 models trained on these datasets. Table 1 reports the list of datasets and the FID values (Heusel et al., 2017) of the source checkpoints. We also experimented with four smaller datasets to verify our conclusions in the medium-shot and few-shot regimes. The details on the datasets are provided in the appendix.

**Experimental setup.** Here, we describe the details of our experimental protocol for both the pretraining of the source checkpoints and the subsequent training on the target datasets. We always use the official PyTorch implementation of StyleGAN2-ADA (Karras et al., 2020a) provided by the authors[1]. We use the "stylegan2" configuration in the ADA implementation with the default hyperparameters (same for all datasets). Training is performed on eight Tesla V100 GPUs and takes approximately three hours per 1M real images shown to the discriminator.

**Pretraining of source checkpoints.** We pretrain one checkpoint on the Imagenet for 50M real images shown to the discriminator and seven checkpoints on other source datasets from Table 1 for 25M images. A larger number of optimization steps for the Imagenet is used since this dataset is more challenging and requires more training epochs to converge. For the large LSUN datasets (Cat, Dog, Bedroom), we use $10^6$ first images to preserve memory. For Satellite-Landscapes, we use ADA due to its smaller size. Then, we always use checkpoints with the best FID for further transferring to target datasets for each source dataset.

**Training on target datasets.** For each source checkpoint, we perform transfer learning to all datasets from Table 1. We use the default transfer learning settings from the StyleGAN2-ADA implementation (faster adaptive data augmentation (ADA) adjustment rate, if applicable, and no $G_{ema}$ warmup). ADA is disabled for the datasets containing more than 50K images and enabled for others with default hyperparameters. In these experiments, we train for 25M real images shown

---

[1] https://github.com/NVlabs/stylegan2-ada-pytorch

to the discriminator. Each transfer experiment is performed with three independent runs, and the metrics are reported for the run corresponding to the median best FID (Heusel et al., 2017).

**Metrics.** In the experiments, we evaluate the performance via the four following metrics. (1) **Frechet Inception Distance (FID)** (Heusel et al., 2017), which quantifies the discrepancy between the distributions of real and fake images, represented by deep embeddings. Both distributions are approximated by Gaussians, and the Wasserstein distance between them is computed. (2) **Precision** (Kynkäänniemi et al., 2019), which measures the realism of fake images, assuming that the visual quality of a particular fake is high if it belongs to the neighborhood of some real images in the embedding space. (3) **Recall** (Kynkäänniemi et al., 2019), which quantifies GAN diversity, measuring the rate of real images that belong to the neighborhood of some fake images in the embedding space. (4) **Convergence rate** equals a number of real images that were shown to the discriminator at the moment when the generator FID for the first time exceeded the optimal FID by at most $5\%$. Intuitively, this metric quantifies how fast the learning process reaches a plateau. FID is computed based on the image embeddings extracted by the InceptionV3 model[2]. Precision and Recall use the embeddings provided by the VGG-16 model[3]. Precision and Recall are al-

| Dataset | Number of images | FID |
|---|---|---|
| Datasets for pretraining | | |
| Imagenet | 1 281 137 | 49.8 |
| LSUN-Cat | 1 000 000 | 7.8 |
| LSUN-Dog | 1 000 000 | 15.0 |
| LSUN-Bedroom | 1 000 000 | 3.3 |
| LSUN-Church | 126 227 | 3.2 |
| Satellite-Landscapes | 2 608 | 26.6 |
| Satellite-Buildings | 280 741 | 12.4 |
| FFHQ | 70 000 | 5.5 |
| Additional target datasets | | |
| CIFAR-10 | 50 000 | — |
| Grumpy Cat | 100 | — |
| Flowers | 8 189 | — |
| Simpsons | 41 866 | — |
| BreCaHAD | 3 253 | — |

Table 1: The datasets used in our experiments. All images are resized to $256 \times 256$ resolution. The last column reports the FID values of the source checkpoints trained with the random initialization.

ways computed with $k{=}5$ neighbors. For FID calculation, we always use all real images and 50K generated samples. For Precision/Recall calculation, we use the first 200K real images (or less, if the real dataset is smaller) and 50K generated samples.

**Results.** The metric values for all datasets are reported in Table 2, where each cell corresponds to a particular source-target pair. For the best (in terms of FID) checkpoint obtained for each source-target transfer, we report the FID value (top row in each cell), Precision and Recall (the second and the third rows in each cell), and the convergence rate measured in millions of images (bottom row in each cell). We highlight the sources that provide the best FID for each target dataset or differ from the best one by at most $5\%$. We additionally present the curves of FID, Precision, and Recall values for several target datasets on Figure 9 and Figure 10 in the appendix.

We describe the key observations from Table 2 below:

- In terms of FID, a pretraining based on a diverse source (e.g., Imagenet or LSUN Dog) is superior to training from scratch on all datasets in our experiments.

- The choice of the source checkpoint significantly influences the coverage of the finetuned model, and the Recall values vary considerably for different sources, especially for smaller target datasets. For instance, on the Flowers dataset, their variability exceeds ten percent. In contrast, the Precision values are less affected by pretraining, and their typical variability is about $2-3\%$. Figure 4 reports the standard deviations of Precision/Recall computed over different sources and highlights that Recall has higher variability compared to Precision, despite the latter having higher absolute values.

- Pretraining considerably speeds up the optimization compared to the training from scratch.

---

[2]https://nvlabs-fi-cdn.nvidia.com/stylegan2-ada-pytorch/pretrained/metrics/inception-2015-12-05.pt

[3]https://nvlabs-fi-cdn.nvidia.com/stylegan2-ada-pytorch/pretrained/metrics/vgg16.pt

| Target | | FFHQ | L.Bedroom | L.Cat | L.Church | L.Dog | S.Buildings | S.Landscapes | Imagenet | From scratch |
|---|---|---|---|---|---|---|---|---|---|---|
| FFHQ | F | | 5.7 | 5.52 | 6.08 | 5.18 | 8.49 | 6.57 | 4.87 | 5.52 |
| | P | | 0.782 | 0.793 | 0.767 | 0.793 | 0.798 | 0.797 | 0.784 | 0.795 |
| | R | | 0.417 | 0.419 | 0.455 | 0.459 | 0.318 | 0.355 | 0.457 | 0.472 |
| | C | | 5 | 8 | 7 | 11 | 14 | 16 | 12 | 13 |
| L.Bedroom | F | 2.71 | | 3.01 | 2.8 | 2.63 | 3.77 | 4.3 | 2.57 | 3.3 ±0.2 |
| | P | 0.664 | | 0.651 | 0.663 | 0.657 | 0.667 | 0.643 | 0.679 | 0.668 |
| | R | 0.469 | | 0.447 | 0.485 | 0.471 | 0.314 | 0.344 | 0.475 | 0.459 |
| | C | 24 | | 25 | 25 | 25 | 21 | 24 | 25 | 23 |
| L.Cat | F | 7.62 | 7.68 | | 7.72 | 6.65 | 9.09 | 9.95 | 7.12 | 7.85 |
| | P | 0.68 | 0.68 | | 0.669 | 0.687 | 0.703 | 0.67 | 0.688 | 0.684 |
| | R | 0.402 | 0.381 | | 0.402 | 0.409 | 0.273 | 0.303 | 0.39 | 0.368 |
| | C | 22 | 25 | | 25 | 18 | 25 | 25 | 21 | 20 |
| L.Church | F | 3.09 | 3.11 | 3.28 | | 2.97 | 3.99 | 6.79 | 3.0 | 3.16 |
| | P | 0.689 | 0.7 | 0.69 | | 0.677 | 0.692 | 0.63 | 0.699 | 0.682 |
| | R | 0.54 | 0.497 | 0.496 | | 0.543 | 0.414 | 0.322 | 0.528 | 0.554 |
| | C | 23 | 22 | 23 | | 21 | 22 | 5 | 23 | 25 |
| L.Dog | F | 14.8 | 15.1 | 13.9 | 15.6 | | 18.4 | 18.4 | 14.4 | 15.02 |
| | P | 0.74 | 0.747 | 0.757 | 0.738 | | 0.753 | 0.754 | 0.743 | 0.758 |
| | R | 0.363 | 0.334 | 0.359 | 0.36 | | 0.237 | 0.256 | 0.365 | 0.349 |
| | C | 22 | 22 | 24 | 25 | | 25 | 25 | 24 | 24 |
| S.Buildings | F | 11.1 | 11.5 | 11.77 | 11.2 | 12.1 | | 16.7 | 10.72 | 12.36 |
| | P | 0.347 | 0.337 | 0.316 | 0.34 | 0.333 | | 0.281 | 0.319 | 0.348 |
| | R | 0.549 | 0.53 | 0.52 | 0.555 | 0.526 | | 0.413 | 0.574 | 0.507 |
| | C | 24 | 20 | 21 | 16 | 25 | | 25 | 25 | 19 |
| S.Landscapes | F | 25.3 | 26.1 | 24.3 | 25.3 | 23.8 | 28.2 | | 21.0 | 26.6 |
| | P | 0.756 | 0.762 | 0.762 | 0.73 | 0.759 | 0.769 | | 0.719 | 0.737 |
| | R | 0.249 | 0.191 | 0.2 | 0.291 | 0.282 | 0.136 | | 0.393 | 0.214 |
| | C | 23 | 21 | 18 | 5 | 8 | 14 | | 2 | 25 |
| CIFAR-10 | F | 8.6 ±0.5 | 8.29 | 7.6 | 8.62 | 7.11 | 10.4 | 9.22 | 6.2 ±0.5 | 9.33 |
| | P | 0.79 | 0.764 | 0.75 | 0.759 | 0.769 | 0.783 | 0.759 | 0.761 | 0.781 |
| | R | 0.493 | 0.45 | 0.479 | 0.502 | 0.525 | 0.397 | 0.401 | 0.559 | 0.455 |
| | C | 15 | 11 | 5 | 8 | 8 | 25 | 19 | 3 | 20 |
| Flowers | F | 9.47 | 9.79 | 9.4 | 9.88 | 8.88 | 11.8 | 9.07 | 8.31 | 10.73 |
| | P | 0.786 | 0.776 | 0.795 | 0.773 | 0.773 | 0.772 | 0.809 | 0.773 | 0.78 |
| | R | 0.251 | 0.215 | 0.194 | 0.226 | 0.269 | 0.14 | 0.153 | 0.282 | 0.271 |
| | C | 22 | 16 | 15 | 25 | 6 | 20 | 21 | 9 | 21 |
| Grumpy Cat | F | 11.9 | 12.3 | 14.0 | 16.1 | 12.6 | 28.5 | 16.3 | 14.7 | 15.34 |
| | P | 0.999 | 0.996 | 0.997 | 0.995 | 0.998 | 0.868 | 0.999 | 0.999 | 0.997 |
| | R | 0.06 | 0.02 | 0.03 | 0.0217 | 0.04 | 0.01 | 0.045 | 0.05 | 0.0175 |
| | C | 24 | 25 | 24 | 25 | 25 | 1 | 25 | 25 | 25 |
| Simpsons | F | 7.87 | 8.0 ±0.4 | 8.12 | 7.93 | 7.67 | 10.0 | 9.98 | 8.28 | 8.42 |
| | P | 0.432 | 0.423 | 0.431 | 0.436 | 0.416 | 0.404 | 0.418 | 0.427 | 0.42 |
| | R | 0.406 | 0.349 | 0.353 | 0.384 | 0.395 | 0.219 | 0.173 | 0.364 | 0.335 |
| | C | 24 | 25 | 24 | 21 | 22 | 24 | 25 | 25 | 25 |
| BreCaHAD | F | 26.31 | 23.12 | 24.80 | 25.40 | 25.36 | 23.81 | 21.84 | 22.73 | 23.72 |
| | P | 0.694 | 0.679 | 0.696 | 0.709 | 0.692 | 0.730 | 0.712 | 0.703 | 0.705 |
| | R | 0.385 | 0.417 | 0.462 | 0.412 | 0.439 | 0.337 | 0.473 | 0.483 | 0.434 |
| | C | 3 | 1 | 1 | 4 | 1 | 2 | 3 | 1 | 11 |

Table 2: Metrics computed for the best-FID checkpoint for different source and target datasets. Each row corresponds to a particular target dataset, and each column corresponds to a particular source model used to initialize the training. For each target dataset, we highlight (by orange) the sources that provide the smallest FID or which FID differs from the best one by at most $5\%$. In each cell, we report from to bottom: **FID**, *Precision*, *Recall*, and convergence rate measured in millions of images (lower is better). In purpose to make the table easier to read, we report std only once it exceeds $5\%$ which happens rarely. The typical values vary around $0.1$.

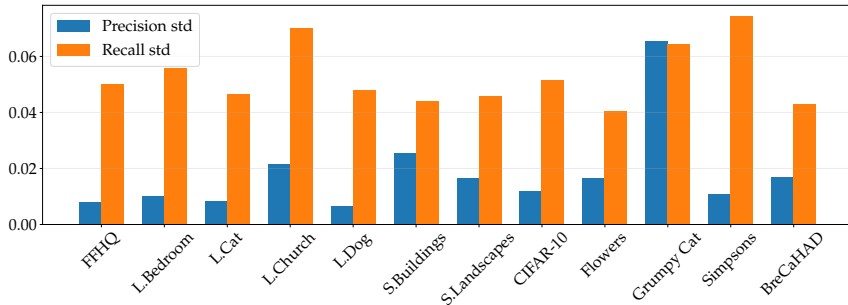

Figure 4: Standard deviations of Precision/Recall values for each target dataset computed over different sources. Due to the symmetric nature of the quantities, the table reports the standard deviation for precision and recall computed over an equal number of real and generated samples (minimum between a dataset size and 50K)

Overall, despite having poor quality (FID=49.8), the Imagenet-pretrained unconditional StyleGAN2 model appears to be a superior GAN initialization that typically leads to more efficient optimization compared to alternatives. This result contradicts the observations in (Wang et al., 2018b) showing that it is beneficial to transfer from dense and less diverse sources rather than sparse and diverse ones, like Imagenet. We attribute this inconsistency to the fact that (Wang et al., 2018b) experimented with the WGAN-GP models, which are significantly inferior to the current state-of-the-art ones.

## 3.2 ANALYSIS

In this section, we perform several additional experiments that illustrate the benefits of pretraining.

**Pretraining improves the mode coverage for real data.** Here, we consider the Flowers dataset and assume that each of its 102 labeled classes corresponds to different distribution modes. To assign the generator samples to the closest mode, we train a 102-way flowers classifier via finetuning the linear head of the Imagenet-pretrained ResNet-50 on real labeled images from Flowers. Then we apply this classifier to generated images from eleven consecutive generator snapshots from the GAN training process on the interval from 0 to 200 kimgs taken every 20 kimgs. This pipeline allows for tracking the number of covered and missed modes during the training process. Figure 5 (left) demonstrates how the number of covered modes changes when GAN is trained from scratch or the checkpoints pretrained on FFHQ and Imagenet. In this experiment, we consider a mode being "covered" if it contains at least ten samples from the generated dataset of size 10 000. One can see that Imagenet, being the most diverse source, initially covers more modes of the target data and faster discovers the others. FFHQ also provides coverage improvement but misses more modes compared to Imagenet even after training for 200 kimgs. With random initialization, the training process covers only a third of modes after training for 200 kimgs . On the right of Figure 5, we show samples drawn for the mode, which is poorly covered by the GAN trained from FFHQ initialization and well-covered by its Imagenet counterpart.

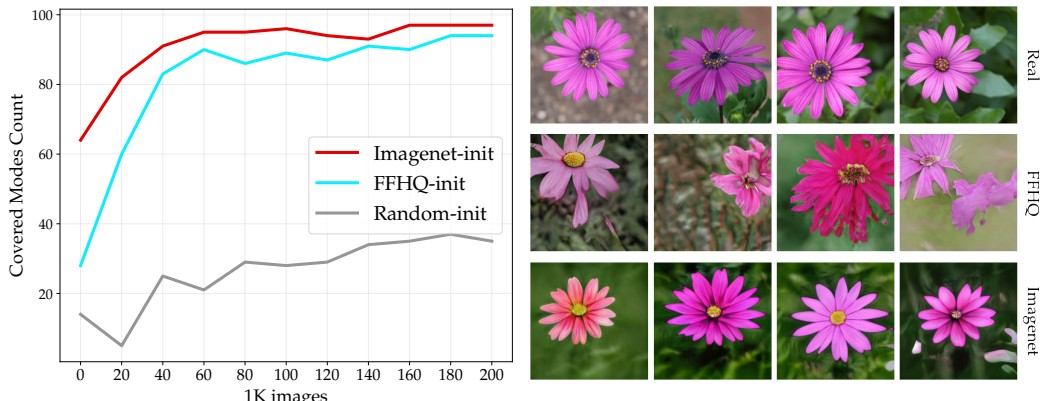

Figure 5: *Left:* Number of modes covered by the generator snapshots during the training process from three different initializations. *Right:* samples of the 65-th class of the Flowers dataset, which is well-covered by the GAN trained from the Imagenet initialization and poorly covered by the GAN trained from the FFHQ initialization. *Top:* real images; *Middle:* FFHQ; *Bottom:* Imagenet.

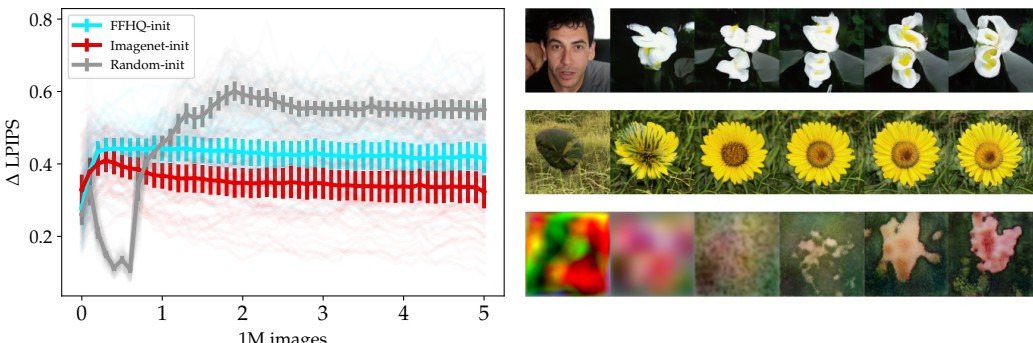

Figure 6: Evolution of the generated samples with different source initializations. *Left*: average LPIPS-distance between images generated by the consecutive generator snapshots for the same latent code. *Right*: images generated with the same latent code evolving during training: *top row*: start from FFHQ, *middle row*: start from Imagenet, *bottom row*: start from random initialization.

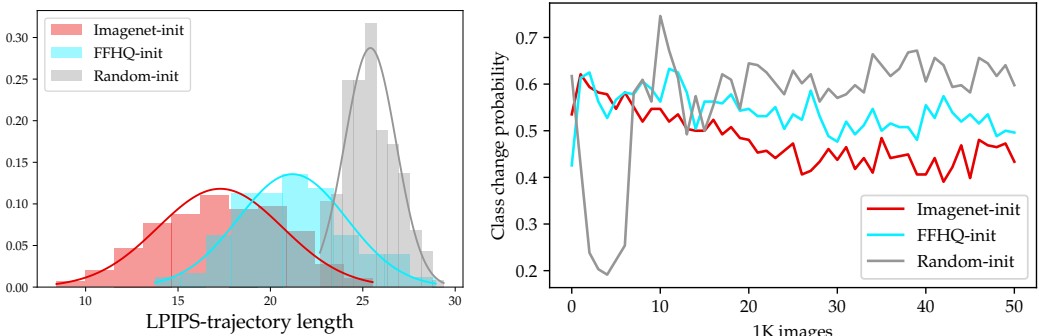

Figure 7: *Left*: distribution of samples trajectories lengths with Flowers as a target dataset. *Right*: generated class change probability for individual latents during the training.

**Pretraining provides more gradual image evolution.** The observations above imply that it is beneficial to initialize training by the checkpoint with a higher recall so that the target data is originally better "covered" by the source model. We conjecture that transferring from a model with higher recall makes it easier to cover separate modes in the target distribution since, in this case, generated samples can slowly drift to the closest samples of the target domain without abrupt changes to cover previously missing modes. To validate this intuition, we consider a fixed batch of $64$ random latent codes $z$ and a sequence of the generator states $G_1, \ldots, G_N$ obtained during the training. Then we quantify the difference between consecutive images computed as the perceptual LPIPS distance Zhang et al. (2018) LPIPS$(G_i(z), G_{i+1}(z))$. Figure 6 shows the dynamics of the distances for Flowers as the target dataset and Imagenet, FFHQ, and random initializations. Since the Imagenet source initially has higher coverage of the target data, its samples need to transform less, which results in higher performance and faster convergence.

Figure 6 indicates more gradual sample evolution when GAN training starts from a pretraining checkpoint. Here we additionally report the distributions of samples' trajectories' lengths quantified by LPIPS. Namely, for a fixed $z$ and a sequence of the generator snapshots $G_1, \ldots, G_N$ obtained during training, we calculate the length of the trajectory as a sum $\sum_i$ LPIPS$(G_i(z), G_{i+1}(z))$. Figure 7 (left) presents the length distributions for three initializations and Flowers as the target dataset.

Finally, to track the dynamics of mode coverage of the target dataset, we obtain the class assignments of the generated samples $G_1(z), \ldots, G_N(z)$ with a classifier pretrained on the Flowers dataset. Then for the samples $G_i(z), G_{i+1}(z)$ generated with the consequent checkpoints, we calculate the probability that the sample changes its class assignment by averaging over $256$ latent codes. That is, we evaluate the probability that a flower class of a sample $G_i(z)$ differs from a class of a sample $G_{i+1}(z)$. The probabilities of the class change for different source checkpoints are presented in Figure 7, right. Importantly, training from pretrained sources demonstrates higher class persistence

| GAN Model (source, target) | Inverted Domain | Best FID | LPIPS-**error** | $F$-**error** |
|---|---|---|---|---|
| s: Random, t: FFHQ | CelebA-HQ | 5.35 | 0.22 | 0.186 |
| s: Imagenet, t: FFHQ | CelebA-HQ | 4.86 | 0.22 | **0.174** |
| s: Random, t: FFHQ | FFHQ | 5.35 | 0.25 | 0.180 |
| s: Imagenet, t: FFHQ | FFHQ | 4.86 | 0.25 | **0.168** |
| s: Random, t: L.Bedroom | L.Bedroom | 2.97 | 0.47 | 0.123 |
| s: Imagenet, t: L.Bedroom | L.Bedroom | 2.56 | **0.44** | **0.115** |

Table 3: Reconstruction errors for GAN models with different source and target datasets.

of individual samples. This indicates that the Imagenet-pretrained generator initially covers the target dataset well enough and requires fewer mode-changing sample hops during training.

**Pretraining is beneficial for downstream tasks.** Here, we focus on the task of inverting a real image given a pretrained generator, which is necessary for semantic editing. We employ the recent GAN inversion approach (Tov et al., 2021) and train the encoders that map real images into the latent space of generators approximating FFHQ and Bedroom distributions. For both FFHQ and Bedroom, we consider the best generators that were trained from scratch and the Imagenet-based pretraining. Table 3 reports the reconstruction errors quantified by the LPIPS measure and $F$-**error**. The details of the metrics computation are provided in the appendix. Overall, Table 3 confirms that higher GAN recall provided by pretraining allows for the more accurate inversion of real images.

## 4   CHOOSING PROPER PRETRAINED CHECKPOINT

This section describes a simple recipe to select the most appropriate pretrained checkpoint to initialize GAN training for a particular target dataset. To this end, we consider a set of natural proxy metrics that quantify the similarity between two distributions. Each metric is computed in two regimes. In the first regime, we measure the distance between the source dataset, consisting of real images used to pretrain the GAN checkpoint, and the target dataset of real images. In the second regime, we use the generated images from pretrained checkpoints instead of the source dataset. The second regime is more practical since it does not require the source dataset. As natural proxy metrics, we consider **FID**, **KID** (Bińkowski et al., 2018), **Precision**, and **Recall** measures.

To estimate the reliability of each metric, we calculate the number of target datasets for which this metric does not correctly predict the optimal starting checkpoint. We consider a starting checkpoint optimal if it provides the lowest FID score or its FID score differs from the lowest by most 5%. The quality for all metrics is presented in Table 4, which shows that FID or Recall can be used as a rough guide to select a pretrained source in both regimes. On the other hand, Precision is entirely unreliable.

| Regime/Metric | FID | KID | Precision | Recall |
|---|---|---|---|---|
| Real Source | 3 | 5 | 11 | **2** |
| Generated Source | 3 | 3 | 7 | 3 |

Table 4: The number of target datasets for which the metrics fail to identify the best source (with up to 5% best FID deviation).

This observation is consistent with our findings from Section 2 that imply that Recall can serve as a predictive measure of finetuning quality.

## 5   CONCLUSION

Transferring pretrained models to new datasets and tasks is a workhorse of modern ML. In this paper, we investigate its success in the context of GAN finetuning. First, we demonstrate that transfer from pretrained checkpoints can improve the model coverage, which is crucial for GANs exhibiting mode-seeking behavior. Second, we explain that it is beneficial to use both pretrained generators and discriminators for optimal finetuning performance. This implies that the GAN studies should open-source discriminator checkpoints as well rather than the generators only. Finally, we show that the recall measure can guide the choice of a checkpoint for transfer and highlight the advantages of Imagenet-based pretraining, which is not currently common in the GAN community. We open-source the StyleGAN2 checkpoints pretrained on the Imagenet of different resolutions for reuse in future research.

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

## 6 APPENDIX

### DATASETS

Here we provide the details for the used datasets. Table 5 reports the size, original image resolution (which was always resized to $256 \times 256$ in our experiments), number of samples used for training, and URL for each of the datasets. In Tables 6, 7, 8, 9 we report pairwise distances between source and target datasets for different metrics. Figure 8 illustrates samples from each dataset. As for BreCaHAD, we generate a dataset of $256 \times 256$ crops of the original dataset images with the code provided in StyleGAN-ADA repository.

| Dataset | Size | Original Resolution | Samples Used |
|---|---|---|---|
| CIFAR-10[4] | 50 000 | $32 \times 32$ | 50 000 |
| FFHQ[5] | 70 000 | $1024 \times 1024$ | 70 000 |
| Flowers[6] | 8 189 | varies | 8 189 |
| Grumpy-Cat[7] | 100 | $256 \times 256$ | 100 |
| Imagenet[8] | 1 281 137 | varies | 1 281 137 |
| LSUN Bedroom[9] | 3 033 042 | $256 \times 256$ | 1 000 000 |
| LSUN Cat[9] | 1 657 266 | $256 \times 256$ | 1 000 000 |
| LSUN Church[9] | 126 227 | $256 \times 256$ | 126 227 |
| LSUN Dog[9] | 5 054 817 | $256 \times 256$ | 1 000 000 |
| Satellite-Buildings[10] | 280 741 | $300 \times 300$ | 280 741 |
| Satellite-Landscapes[11] | 2 608 | $1800 \times 1200$ | 2 608 |
| Simpsons[12] | 41 866 | varies | 41 866 |
| BreCaHAD[13] | 3 253 | $256 \times 256$ | 3 253 |

Table 5: Datasets information.

### LEARNING CURVES

On Figure 9 and Figure 10 we present the learning curves from Table 2 in the main text. To make the plots readable, for each target dataset, we report only the curves corresponding to training from scratch, training from the Imagenet checkpoint, and from two checkpoints that perform best among the rest as a representative subset of sources.

### SYNTHETIC DATA DETAILS

Here we provide the details for the experiment described in Section 2.2. The synthetic target data is formed by 10 Gaussians with centers on the circle of radius 20 and $\sigma = 0.25$. Source-I (blue) is a distribution formed as a sum of a uniform distribution on a zero-centered circle of a radius 20 and the zero-centered Gaussians with $\sigma = 4$. Source-II (green) is formed by 3 Gaussians with centers that coincide with the consequent centers of three Gaussians of the original data and $\sigma = 0.5$. We use the standard GAN loss (Goodfellow et al., 2014) and perform 5000 generator training steps with 4 discriminator steps for every generator step. We use batch size 64 and Adam optimizers with learning rate 0.0002 and $\beta_1, \beta_2 = 0.5, 0.999$. The generator has a 64-dimensional latent space

---

[4]https://www.cs.toronto.edu/~kriz/cifar.html
[5]https://github.com/NVlabs/ffhq-dataset
[6]https://www.robots.ox.ac.uk/~vgg/data/flowers/102/index.html
[7]https://hanlab.mit.edu/projects/data-efficient-gans/datasets/
[8]https://image-net.org/index.php
[9]https://www.yf.io/p/lsun
[10]https://www.aicrowd.com/challenges/mapping-challenge-old
[11]https://earthview.withgoogle.com
[12]https://www.kaggle.com/c/cmx-simpsons/data
[13]https://figshare.com/articles/dataset/BreCaHAD_A_Dataset_for_Breast_Cancer_Histopathological_Annotation_and_Diagnosis/7379186

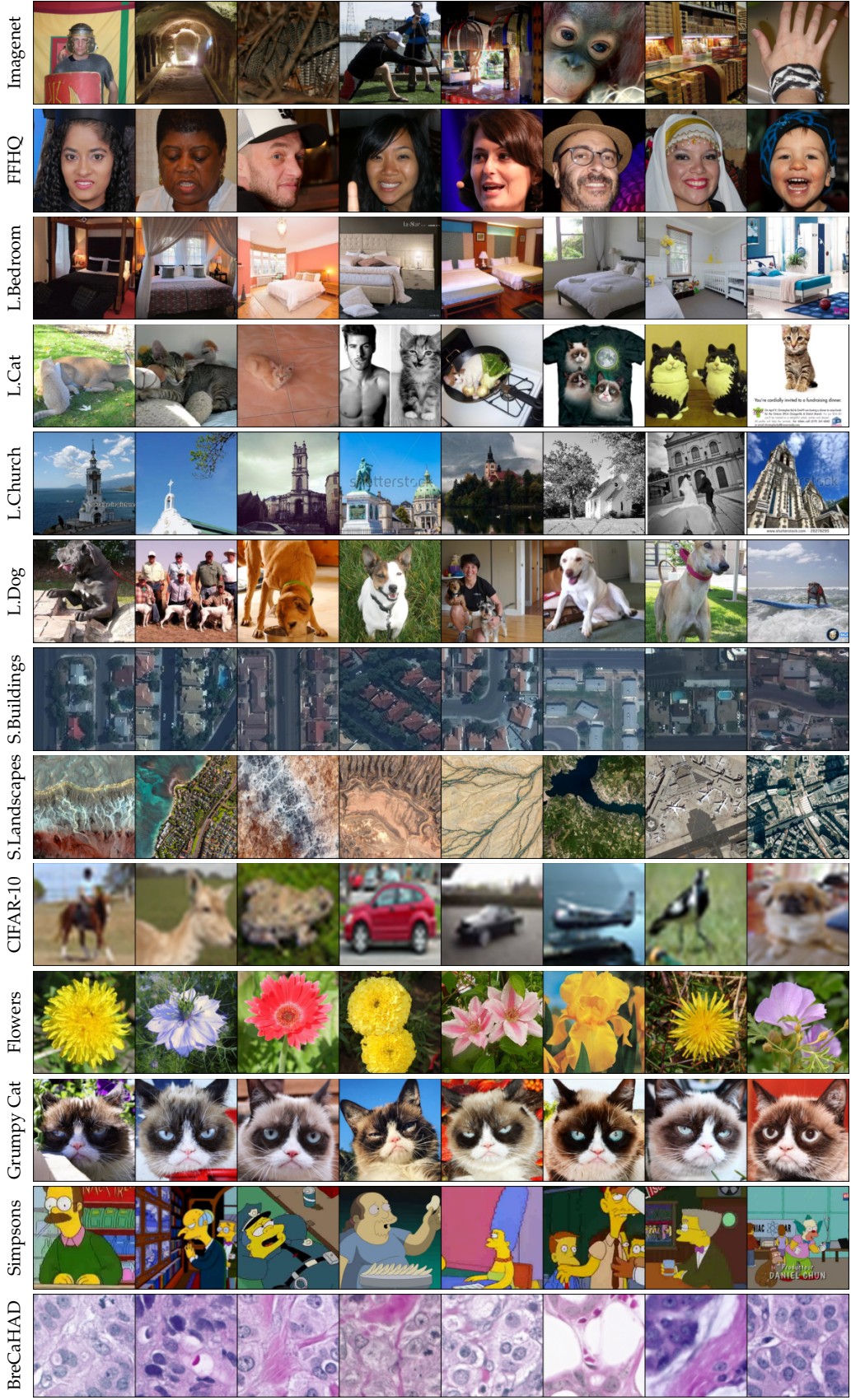

Figure 8: Samples for each of the target and source datasets.

|      | F     | L.B   | L.Ca  | L.Ch  | L.Dog | S.B   | S.L   | I     |
|------|-------|-------|-------|-------|-------|-------|-------|-------|
| F    | 0     | 244.0 | 194.8 | 240.5 | 178.8 | 256.1 | 233.3 | 150.8 |
| L.B  | 244.0 | 0     | 165.0 | 182.8 | 162.4 | 233.3 | 236.7 | 143.4 |
| L.Ca | 194.8 | 165.0 | 0     | 200.8 | 97.6  | 206.9 | 185.9 | 104.1 |
| L.Ch | 240.5 | 182.8 | 200.8 | 0     | 167.0 | 199.8 | 232.5 | 140.4 |
| L.Dog| 178.8 | 162.4 | 97.6  | 167.0 | 0     | 200.0 | 182.3 | 63.9  |
| S.B  | 256.1 | 233.3 | 206.9 | 199.8 | 200.0 | 0     | 172.2 | 177.5 |
| S.L  | 233.3 | 236.7 | 185.9 | 232.5 | 182.3 | 172.2 | 0     | 145.3 |
| C    | 197.2 | 188.1 | 120.9 | 192.3 | 102.2 | 202.1 | 185.3 | 85.4  |
| Fl   | 257.7 | 254.7 | 235.4 | 243.8 | 215.9 | 285.4 | 261.4 | 192.8 |
| GC   | 293.1 | 260.8 | 188.4 | 259.2 | 259.3 | 341.4 | 334.5 | 264.4 |
| S    | 252.5 | 225.2 | 199.4 | 218.8 | 195.9 | 217.7 | 244.3 | 167.6 |
| BCH  | 347.8 | 345.7 | 319.7 | 356.0 | 303.8 | 351.2 | 245.4 | 280.4 |

Table 6: FID distances between source and target datasets. Underlined cell in a row corresponds to a source domain that is closest to a fixed target. Datasets names are shortened as: **L.Bdr** (LSUN Bedroom), **L.Cat** (LSUN Cat), **L.Chr** (LSUN Church), **L.Dog** (LSUN Dog), **S.Bld** (Satellite Buildings), **S.Lnd** (Satellite Landscapes), **Imgn** (Imagenet), **C-10** (CIFAR-10), **Flw** (Flowers), **GC** (Grumpy Cat), **S** (Simpsons), **BCH** (BreCaHAD).

|      | F     | L.B   | L.Ca  | L.Ch  | L.Dog | S.B   | S.L   | I     |
|------|-------|-------|-------|-------|-------|-------|-------|-------|
| F    | 0     | 0.237 | 0.169 | 0.213 | 0.116 | 0.230 | 0.165 | 0.116 |
| L.B  | 0.237 | 0     | 0.161 | 0.193 | 0.124 | 0.249 | 0.200 | 0.126 |
| L.Ca | 0.168 | 0.161 | 0     | 0.185 | 0.080 | 0.189 | 0.129 | 0.105 |
| L.Ch | 0.213 | 0.193 | 0.185 | 0     | 0.114 | 0.202 | 0.185 | 0.096 |
| L.Dog| 0.116 | 0.125 | 0.079 | 0.113 | 0     | 0.155 | 0.095 | 0.027 |
| S.B  | 0.229 | 0.248 | 0.189 | 0.202 | 0.156 | 0     | 0.129 | 0.179 |
| S.L  | 0.165 | 0.200 | 0.130 | 0.185 | 0.095 | 0.129 | 0     | 0.109 |
| C    | 0.137 | 0.149 | 0.092 | 0.144 | 0.048 | 0.170 | 0.117 | 0.060 |
| Fl   | 0.227 | 0.260 | 0.211 | 0.230 | 0.157 | 0.277 | 0.212 | 0.153 |
| GC   | 0.260 | 0.283 | 0.113 | 0.276 | 0.196 | 0.332 | 0.249 | 0.195 |
| S    | 0.265 | 0.276 | 0.215 | 0.244 | 0.178 | 0.247 | 0.227 | 0.179 |
| BCH  | 0.335 | 0.374 | 0.316 | 0.375 | 0.267 | 0.349 | 0.205 | 0.273 |

Table 7: KID distances between source and target datasets computed. Highlighted cell in a row corresponds to a source domain that is closest to a fixed target.

and consists of six consequent linear layers, all but the last followed by batch-norms and ReLU-activations. The intermediate layers' sizes are $64, 128, 128, 128, 64$. The discriminator is formed by a sequence of five linear layers, each but the last followed by the ReLU-activation. The intermediate layers' sizes are $64, 128, 128, 64$.

The starting checkpoints for the Dissecting Contributions experiments are taken from the intermediate checkpoints of the GAN training for Source-I. We take every 50-th checkpoint, gathering 100 in total. We perform fine-tuning to the target distribution with the same parameters as above except the number of steps equals 1000.

## LONGER TRAINING

In this series of experiments, we run GAN training for a source checkpoint being either Imagenet-pretrained or randomly initialized for two times higher number of steps (50 million real images shown to the discriminator). The results are presented in Table 10. Generally, Imagenet-pretraining almost always either improves GAN quality or performs equally to the random initialization while speeding up convergence by a large margin.

|  | F | L.B | L.Ca | L.Ch | L.Dog | S.B | S.L | I |
|---|---|---|---|---|---|---|---|---|
| F | 1 | 0.000 | 0.014 | 0.000 | 0.057 | 0.001 | 0.000 | 0.005 |
| L.B | 0.333 | 1 | 0.333 | 0.235 | 0.337 | 0.307 | 0.058 | 0.021 |
| L.Ca | 0.448 | 0.598 | 1 | 0.253 | 0.384 | 0.619 | 0.229 | 0.094 |
| L.Ch | 0.027 | 0.050 | 0.007 | 1 | 0.058 | 0.208 | 0.016 | 0.003 |
| L.Dog | 0.539 | 0.679 | 0.591 | 0.350 | 1 | 0.726 | 0.265 | 0.144 |
| S.B | 0.000 | 0.000 | 0.000 | 0.000 | 0.000 | 1 | 0.000 | 0.000 |
| S.L | 0.007 | 0.014 | 0.007 | 0.042 | 0.002 | 0.705 | 1 | 0.016 |
| C | 0.000 | 0.000 | 0.000 | 0.000 | 0.000 | 0.001 | 0.000 | 0.000 |
| Fl | 0.006 | 0.000 | 0.001 | 0.000 | 0.000 | 0.002 | 0.012 | 0.003 |
| GC | 0.000 | 0.000 | 0.001 | 0.000 | 0.000 | 0.000 | 0.000 | 0.000 |
| S | 0.000 | 0.000 | 0.000 | 0.012 | 0.000 | 0.046 | 0.000 | 0.000 |
| BCH | 0.000 | 0.000 | 0.000 | 0.000 | 0.000 | 0.000 | 0.000 | 0.000 |

Table 8: The Precision values computed for the targets datasets w.r.t. the source datasets.

|  | F | L.B | L.Ca | L.Ch | L.Dog | S.B | S.L | I |
|---|---|---|---|---|---|---|---|---|
| F | 1 | 0.333 | 0.448 | 0.027 | 0.539 | 0.000 | 0.007 | 0.737 |
| L.B | 0.000 | 1 | 0.598 | 0.050 | 0.679 | 0.000 | 0.014 | 0.124 |
| L.Ca | 0.014 | 0.333 | 1 | 0.007 | 0.591 | 0.000 | 0.007 | 0.218 |
| L.Ch | 0.000 | 0.235 | 0.253 | 1 | 0.350 | 0.000 | 0.042 | 0.303 |
| L.Dog | 0.057 | 0.337 | 0.384 | 0.058 | 1 | 0.000 | 0.002 | 0.325 |
| S.B | 0.001 | 0.307 | 0.619 | 0.208 | 0.726 | 1 | 0.705 | 0.533 |
| S.L | 0.000 | 0.058 | 0.229 | 0.016 | 0.265 | 0.000 | 1 | 0.378 |
| C | 0.001 | 0.053 | 0.240 | 0.006 | 0.340 | 0.000 | 0.003 | 0.718 |
| Fl | 0.001 | 0.183 | 0.249 | 0.010 | 0.410 | 0.000 | 0.017 | 0.708 |
| GC | 0.000 | 0.020 | 0.790 | 0.000 | 0.970 | 0.000 | 0.000 | 0.000 |
| S | 0.013 | 0.324 | 0.328 | 0.060 | 0.379 | 0.000 | 0.045 | 0.294 |
| BCH | 0.001 | 0.203 | 0.428 | 0.053 | 0.546 | 0.006 | 0.553 | 0.789 |

Table 9: The Recall values computed for the targets datasets w.r.t. the source datasets.

| Dataset | From Scratch | | | | Imagenet pretraining | | | |
|---------|----------|------|-----------|--------|----------|------|-----------|--------|
|         | Step (M) | FID  | Precision | Recall | Step (M) | FID  | Precision | Recall |
| L.Bedroom | 50 | 2.50 | 0.663 | 0.485 | 50 | 2.33 | 0.691 | 0.483 |
| L.Cat | 42 | 6.87 | 0.686 | 0.394 | 48 | 6.35 | 0.712 | 0.385 |
| L.Church | 36 | 3.01 | 0.705 | 0.547 | 12 | 3.00 | 0.693 | 0.523 |
| L.Dog | 40 | 12.7 | 0.751 | 0.384 | 45 | 12.8 | 0.753 | 0.382 |
| S.Buildings | 35 | 11.9 | 0.363 | 0.498 | 14 | 10.9 | 0.304 | 0.591 |
| S.Landscapes | 25 | 27.4 | 0.737 | 0.214 | 1 | 21.1 | 0.721 | 0.393 |

Table 10: Number of real images shown to the discriminator (step) for the checkpoint with the best FID value, this value and corresponding precison and recall values for long-term trainings with two initialization options.

| Source Model | FID | Precision | Recall | Steps to Convergance |
|--------------|-----|-----------|--------|----------------------|
| Imagenet | 8.31 | 0.77 | 0.28 | 9 |
| Imagenet (half) | 8.54 | 0.81 | 0.22 | 25 |
| FFHQ | 9.47 | 0.79 | 0.25 | 22 |
| FFHQ (half) | 9.5 | 0.77 | 0.27 | 25 |

Table 11: Finetuning to Flowers from a converged source checkpoint and from a checkpoint that passes two times fewer steps.

## TRANSFER FROM AN EARLIER EPOCH

This experiment verifies if it is important to transfer from a well-converged nearly-optimal source checkpoint, or it is sufficient to start from a roughly stabilized checkpoint from the intermediate step of the optimization process. To address the question, we perform a series of additional experiments with Imagenet and FFHQ as source domains, and Flowers as a target domain. As pretrained checkpoints, we consider the best-FID checkpoint and a checkpoint that passed two times fewer steps. The results for these runs are presented in Table 11. Overall, the choice between two options has only a marginal impact on the transfer quality, and one can use the source checkpoint from the middle of training to initialize the finetuning process.

## DETAILS OF EXPERIMENTS ON THE GAN INVERSION

We take the e4e generator inversion approach proposed by (Tov et al., 2021) and train an encoder that maps real data to the GAN latent space. This scheme is known to be capable of mapping real images to the GAN latent space preserving all generator properties such as latent attributes manipulations. We follow the original author's implementation and train an independent encoder model for each generator. For a generator $G$ we receive an encoder $E$ which is trained to satisfy $G(E(x))=x$ for each real data sample $x$. We evaluate the encoders with the average LPIPS-distance (Zhang et al., 2018) between a test set real samples and their inversions equal $\mathbb{E}_{x \sim p_{\text{test}}} \text{LPIPS}(x, G(E(x)))$. We also report the average distance between an original image and its reconstruction features of a pretrained features extractor $F$ which is equal $\mathbb{E}_{x \sim p_{\text{test}}} \|F(x) - F(G(E(x)))\|_2$. The lower these quantities –, the better reconstruction quality is. Following (Tov et al., 2021), for FFHQ-target generators, we train the encoder on the FFHQ dataset and evaluate it on the Celeba-HQ dataset and on FFHQ itself. As for LSUN-Bedroom, we split the original data into a train and a test subset in the proportion 9 : 1 and train e4e on the train set and evaluate on the test set. As the feature extractor $F$, for FFHQ we use a Face-ID pretrained model, same as in (Tov et al., 2021), and MoCo-v2 (Chen et al., 2020b) model for LSUN-Bedroom.

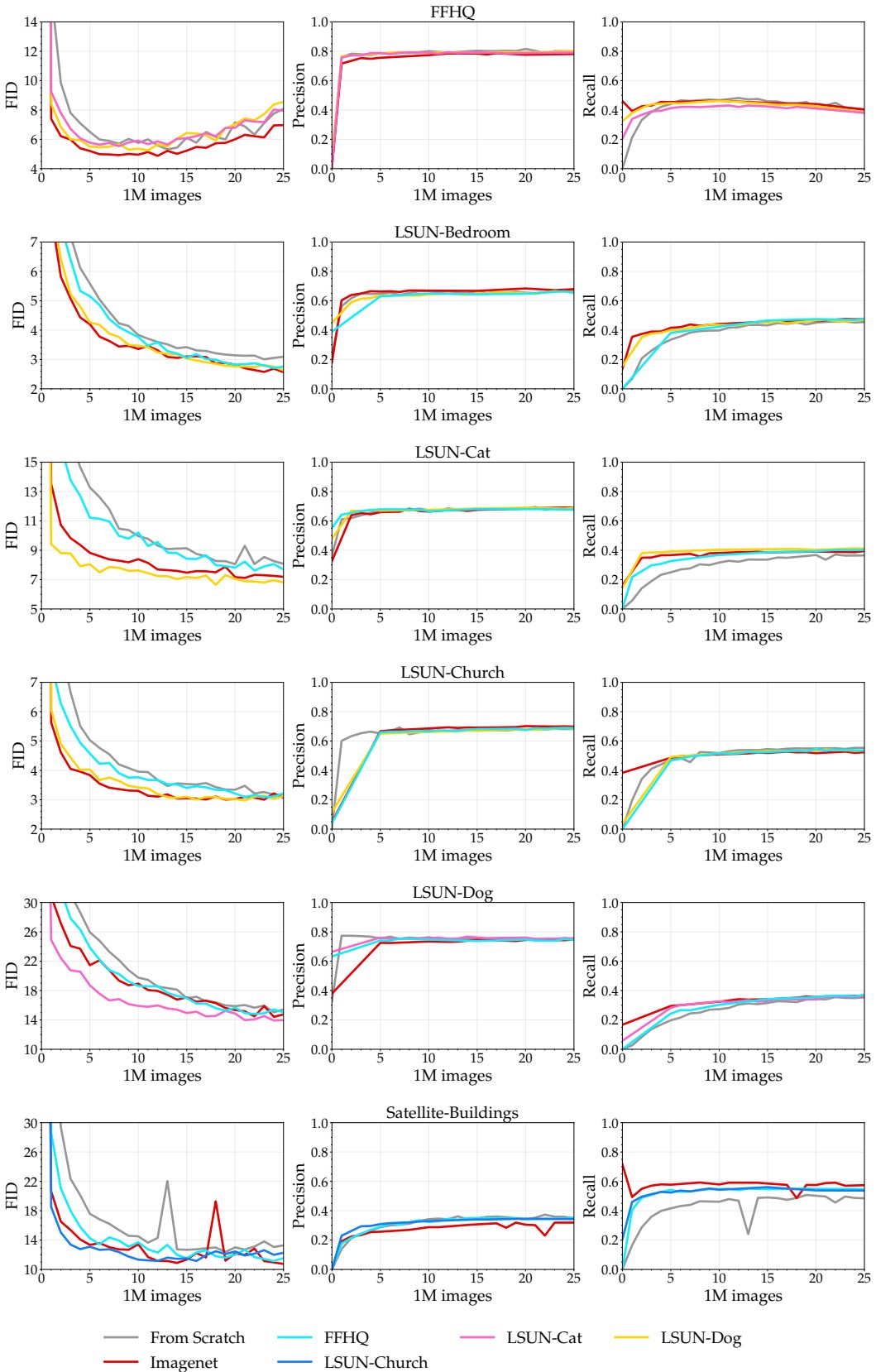

Figure 9: Learning curves for different target and sources datasets, part 1.

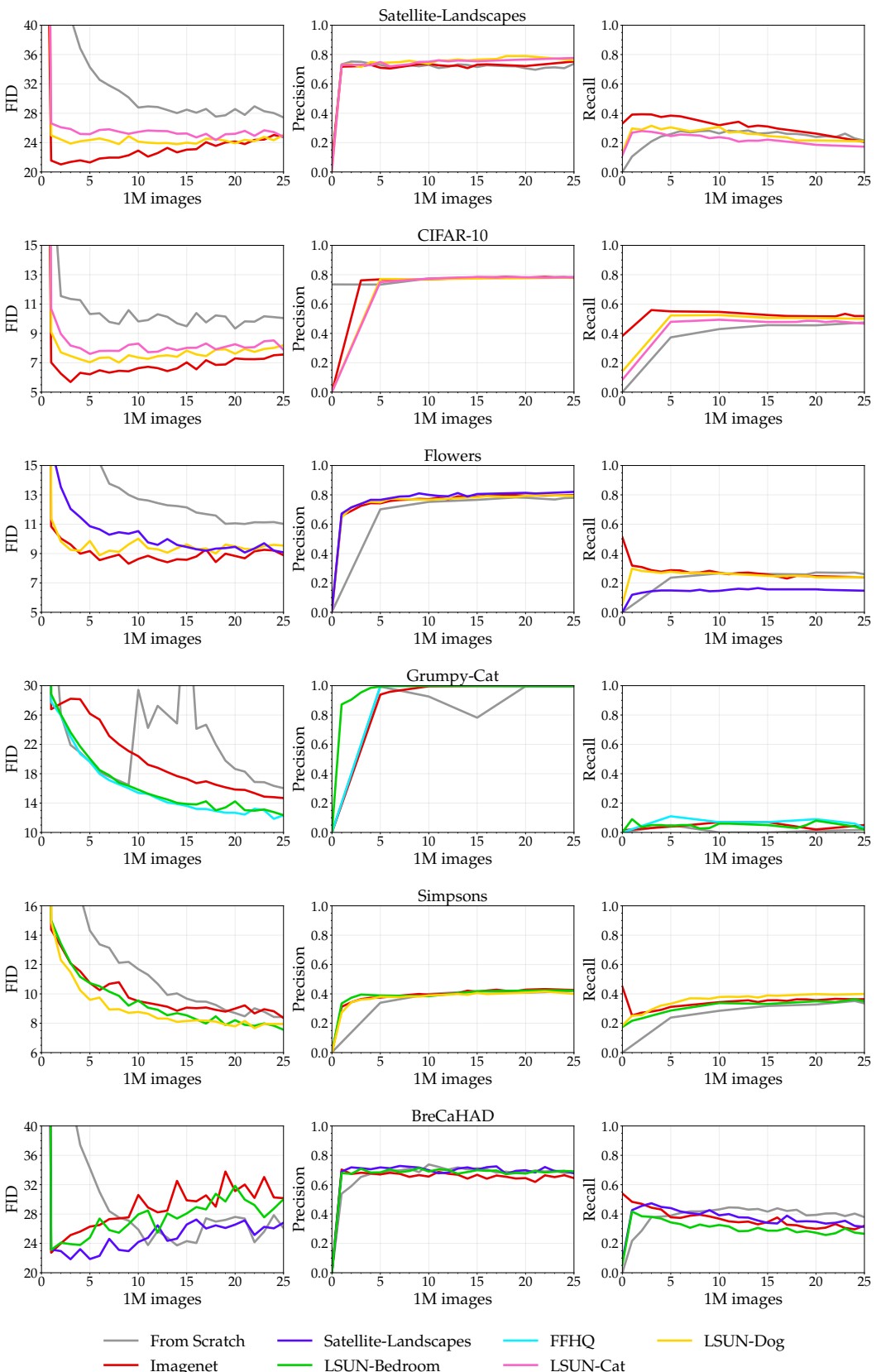

Figure 10: Learning curves for different target and sources datasets, part 2.

