# OpenReview forum: "When, Why, and Which Pretrained GANs Are Useful?"
_ICLR.cc/2022/Conference — ICLR 2022 Poster_

### Official Review · Reviewer_8amM · 2021-11-01

**Correctness:** 3
**Technical Novelty And Significance:** 3
**Empirical Novelty And Significance:** 4
**Recommendation:** 8
**Confidence:** 4

**Main Review:**

While some aspects of the work have been hinted at in prior art, to the best of my knowledge no one has properly studied the questions so eloquently put forward in the title. These are important questions and I for one have wanted to know the answers for some time! The paper does a good job building intuition through toy examples, large-scale transfer test (Table 2), and some quite novel tests (Figs 6 & 7). Overall this feels like important work that should to be accepted. I will champion this paper.

A possible error:
- p.3 second to last para: "... have significant negative correlations with the finetuned GAN quality". I was very surprised to read this and in fact had to backtrack twice. I suspect this statement is strictly false. Unless I'm mistaken, the negative correlation is with Wasserstein distance, but small distance implies high quality, and thus there is a strong POSITIVE correlation with quality. Right?

Some minor complaints/suggestions/questions:
- [citation] is not a noun. It's silent when reading the sentence.
- Personally I found the contribution list at the end of Sec 1 completely unnecessary. We have just read all the same items... (twice if you count the abstract)
- Thumb up/down in Figure 1 may not be the best way to convey the message.
- p5: ImageNet pretraining is always useful: This is an important finding, but LSUN Dog and FFHQ were also uniformly beneficial. ImageNet was just a bit better. Please consider rewording. There is no magic in ImageNet, and it's very likely that some higher quality diverse dataset would almost certainly be even better.
- Table 2: "Random" on the top row was initially mysterious. You mean it's random initialized, while the text up to this point has been talking about training from scratch. Consider a clarification, maybe in the caption.
- Fig 7: What exactly does the class change probability mean? Probability during which timespan? Between init and current, or perhaps current and some previous timestep. (the answer isn't too important, just make sure to be specific).
- Q: The method in Sec 4 is used to choose a particular snapshot. This is nice. Now, if I understand correctly, you're using it for selecting the best snapshot per source dataset -- did you consider using the same heuristic for also selecting the best source dataset/initialization dataset? If it doesn't work, why?




**Summary Of The Paper:**

This paper performs a large-scale study of transfer learning in GANs. It proposes a way to understand the relevance of a pre-trained generator and discriminator, as well as heuristics to select good source/initialization dataset and even a training snapshot. All of this is very valuable for the practitioners.

**Summary Of The Review:**

This paper answers important questions in GAN transfer learning, and therefore makes a solid contribution.

---

> ### Author Response · Authors · 2021-11-22
> **Authors Response**
>
> We thank the reviewer for the positive feedback and answer their questions below.
>
> R: *p.3 second to last para: "... have significant negative correlations with the finetuned GAN quality".
> I was very surprised to read this and in fact had to backtrack twice. I suspect this statement is strictly false.
> Unless I'm mistaken, the negative correlation is with Wasserstein distance, but small distance implies high quality,
> and thus there is a strong POSITIVE correlation with quality. Right?*
>
> Thank you for noticing this sentence that indeed could confuse. We meant that there is a negative correlation between the values i.e. higher recall implies lower W1-distance. We reformulate this sentence in the updated revision.
>
> R: *The method in Sec 4 is used to choose a particular snapshot. This is nice.
> Now, if I understand correctly, you're using it for selecting the best snapshot per source dataset --
> did you consider using the same heuristic for also selecting the best source dataset/initialization dataset? If it doesn't work, why?*
>
> Once a source dataset is chosen, the final transferred model has a neglectable variation w.r.t. a particular source checkpoint (as reported in Table 2, the typical values of the final transferred FID std is around 0.1). Therefore, the primary question of the source choice is how to choose the optimal source domain rather than the optimal checkpoint from this domain. The experiments in Section 4 choose a snapshot among datasets, that is, the number of snapshots is equal to the number of source datasets. We consider two scenarios for the choice of a source for pretraining: based on the source dataset and based on the source checkpoint (lines 1 and 2 in Table 4 correspondingly).
>
> R: *Additional suggestions.*
>
> We thank you for the detailed feedback! Will use it to improve the revision.

---

### Official Review · Reviewer_iExS · 2021-11-02

**Correctness:** 4
**Technical Novelty And Significance:** 2
**Empirical Novelty And Significance:** 3
**Recommendation:** 6
**Confidence:** 3

**Main Review:**

Pros:
(1). This kind of practical paper has significance for the entire community. This paper makes some conclusions, which could promote the development of adapting pre-trained GANs to different domains.
(2). Part 2.1 “high-level intuition” is very interesting.
(3). The writing is clear and easy to understand.

Cons:
(1). It would be better to add a “related works” section to compare this paper with related papers;
(2). In the introduction part, the author claimed that “some conclusions from Wang are not confirmed for modern architectures”. It would be more clear to summarize which conclusion is not confirmed in the related work part.
(3). It would be better to list the distance between source datasets and target datasets in Table 1.


**Summary Of The Paper:**

This paper studies the problem: when, why, and which pretrained GANs are useful.


**Summary Of The Review:**

In general, this paper has conducted a series of experiments with the stylegan2, and got some conclusions, which are helpful for subsequent papers and the community. The analysis and experiments are not very comprehensive. In many ways, [1] is more comprehensive than this one, although [1] is from a few years ago.

[1]. Transferring GANs: generating images from limited data

---

> ### Author Response · Authors · 2021-11-22
> **Authors Response**
>
> We thank the reviewer for their time and address their concerns below.
>
> R: *(1). It would be better to add a “related works” section to compare this paper with related papers; (2). In the introduction part, the author claimed that “some conclusions from Wang are not confirmed for modern architectures”.
> It would be more clear to summarize which conclusion is not confirmed in the related work part.*
>
> Since the number of papers that investigate the process of GAN pretraining is small, we have decided to merge the introduction with an overview of the prior work. Though we discuss the differences with [1] throughout the paper (results subsection in 3.1), we agreed that it would be reasonable to highlight them in the introduction. In the updated paper revision we added the details.
>
> R: *It would be better to list the distance between source datasets and target datasets in Table 1.*
>
> Agreed. To keep the ease of perception of Table 1, we add the correspondent tables in the updated supplementary (Tables 6-9).
>
> [1] Transferring gans: generating images from limited data, Yaxing Wang et. al.

---

### Official Review · Reviewer_BE8y · 2021-11-03

**Correctness:** 4
**Technical Novelty And Significance:** 3
**Empirical Novelty And Significance:** 3
**Recommendation:** 6
**Confidence:** 4

**Main Review:**

**Strengths:**
- Well written and easy to understand.
- Transfer learning for GANs is an important subject. Better pretrained models could save much training time and enable users with less computational resources to achieve good results.
- Simple synthetic experiment gives good insight into the roles that generator and discriminator play during transfer learning.

**Weaknesses:**
- One of the main conclusions of this work is that diverse source datasets (i.e., ImageNet) result in the best performance during transfer learning. However, most of the target datasets contain classes which overlap with ImageNet classes in some form, so it is not surprising that this model transfers well. It would be good to see how ImageNet transfers to datasets it does not overlap much, such as those from the medical imaging domain (one option is the BreCaHAD dataset from [1]).
- An aspect that I think is missing from this study is the level of convergence of the source model. That is to say, is it better to take a checkpoint that it still in the middle of training, or one that has completely converged? What is the optimal time to take a checkpoint?
- No confidence intervals on most experimental results. This is understandable due to the excessive compute budget required to train each of these models, but it still makes it difficult to understand the significance of the results.

**Other Questions and Comments:**
- In Table 2, row Grumpy Cat, column ImageNet, the P and R values seem very off compared to values from other models, and they do not match the values shown in Figure 10.
-In Figure 9, why does ImageNet pretrained models often decrease very fast and then get worse? Is the model collapsing because the discriminator is quickly overfitting? Shouldn't the adaptive data augmentation prevent this kind of catastrophic collapse?
- In Figure 10, FFHQ, why does the model trained from scratch diverge? I would have expected StyleGAN2 to perform very well on FFHQ since this is the dataset it was designed for.
- Many lines are missing from plots in Figure 10 and 11. Why is this?
- If ImageNet checkpoint was not an option to use, would the proposed checkpoint selection method still have selected the best choice of the remaining options?

[1] A. Aksac, D. J. Demetrick, T. Ozyer, and R. Alhajj. BreCaHAD: A dataset for breast cancer histopathological annotation and diagnosis. BMC Research Notes, 12, 2019.

**Summary Of The Paper:**

This paper aims to achieve a better understanding of GAN fine-tuning. A synthetic experiment demonstrates that pretrained discriminators improve the quality of the initial gradients, and pretrained generators help with improving mode coverage. Transfer learning experiments on real image datasets reveal that pretraining primarily improves mode coverage rather than sample fidelity, and that datasets containing a diverse set of images are best suited for transfer.

**Summary Of The Review:**

Paper provides useful insights about the role of the generator and discriminator in GAN transfer learning, as well as the behaviour of transfer learning itself (i.e., improved coverage but not fidelity).

---

> ### Author Response · Authors · 2021-11-22
> **Authors Response (2 / 2)**
>
> R: *Many lines are missing from plots in Figure 10 and 11. Why is this?*
>
> The goal of these plots is to demonstrate representative behavior of training curves for different source and target datasets. To this end, we always report the advocated Imagenet pretraining, random start, and include some representative curves for other source datasets. Since using different source checkpoints corresponds to the learning curves with FID values of different scales, reporting all of them makes the plots hardly readable.
>
> R: *If ImageNet checkpoint was not an option to use,
> would the proposed checkpoint selection method still have selected the best choice of the remaining options?*
>
> Yes. Below we report the results similar to Table 4 in the main text, with Imagenet-source excluded. Each value indicates the number a particular method fails to identify the best starting checkpoint choice.
>
> | Regime \ Metric | FID | KID | Precision | Recall |
> |-----------------|-----|-----|-----------|--------|
> | Real Source     | 3   | 2   | 8         | 2      |
> | Gen. Source     | 4   | 2   | 6         | 3      |
>
> thus the proposed checkpoint choice method remains valid once the Imagenet pretraining is excluded.
>
>
> ------------
>
>
> [1] Training Generative Adversarial Networks with Limited Data, Tero Karras et. al.

---

> ### Author Response · Authors · 2021-11-22
> **Authors Response (1 / 2)**
>
> We thank the reviewer for the feedback and address their questions below.
>
> R: *One of the main conclusions of this work is that diverse source datasets <...> most of the target datasets contain classes which overlap with ImageNet classes in some form <...> (one option is the BreCaHAD dataset from [1])*
>
> Note that, among others, we experiment with the Satellite Landscapes and Satellite Buildings datasets, which, we believe, are significantly different from Imagenet. Nevertheless, we agree that the evaluation on a dataset from the medical imaging domain will be informative. We have performed an additional experiment with the BreCaHAD dataset proposed by the reviewer, the results are provided below.
>
> | Source \ Metric | FID  | Precision | Recall | Steps to Convergance |
> |-----------------|------|-----------|--------|----------------------|
> | Imagenet        | 56.1 | 0.94      | 0.46   | 1                    |
> | LSUN Cat        | 59.4 | 0.95      | 0.30   | 1                    |
> | FFHQ            | 61.5 | 0.95      | 0.27   | 1                    |
> | LSUN Bedroom    | 59.3 | 0.98      | 0.01   | 15                   |
> | Sat. Land.      | 59.8 | 0.93      | 0.20   | 1                    |
> | Sat. Build.     | 57.6 | 0.98      | 0.01   | 11                   |
> | Random          | 78.4 | 0.95      | 0.13   | 2                    |
>
> Overall, the results mirror our observations for other datasets. While the Precision values are less affected by pretraining, the Recall values strongly depend on the choice of the source checkpoint. Specifically, the Imagenet checkpoint demonstrates remarkably higher Recall compared to the random initialization and other models. We will add the BreCaHAD dataset in the final revision when all the runs/sources will be evaluated.
>
> R: *<...> is it better to take a checkpoint that is still in the middle of training, or one that has completely converged?*
>
> To address the question, we perform a series of additional experiments with Flowers as a target domain, and Imagenet/FFHQ as source domains. As pretrained checkpoints, we consider the best-FID checkpoint and a checkpoint that passed two times fewer optimization steps. The results for these runs are presented in the table below.
>
> | Source \ Metric | FID   | Precision | Recall | Steps to Convergance |
> |-----------------|-------|-----------|--------|----------------------|
> | Imagenet        | 8.31  | 0.77      | 0.28   | 9                    |
> | Imagenet (half) | 8.54  | 0.81      | 0.22   | 25                   |
> | FFHQ            | 9.47  | 0.79      | 0.25   | 22                   |
> | FFHQ (half)     | 9.5   | 0.77      | 0.27   | 25                   |
>
>
> Overall, the choice between two options has only a marginal impact on the transfer quality, and one can use the source checkpoint from the middle of training to initialize the optimization process.
>
> R: *No confidence intervals on most experimental results.*
>
> For better readability, Table 2 reports STD values only if they are larger than 5% of the optimal FID value and could potentially affect the sources ranking. Notably, for most source-target pairs these STD values are negligibly small.
>
> R: *In Table 2, row Grumpy Cat, column ImageNet, the P and R values seem very off compared to values from other models,
> and they do not match the values shown in Figure 10.*
>
> We thank the reviewer for pointing out this typo. We fixed it in the updated text revision. The true values are: 14.7 (FID), 0.999 (Precision), 0.05 (Recall), 25 (Convergence rate). The metrics, std values, and rankings were computed with the true values and this is the only place affected by the typo. All the conclusions are valid.
>
> R: *In Figure 9, why does ImageNet pretrained models often decrease very fast and then get worse?*
>
> While the adaptive augmentation partially prevents the overfitting, it is still common for GAN models to deteriorate after a certain optimization step. In particular, a similar behavior was reported in the original StyleGAN2-ADA paper for some of the datasets with limited data ([1], Figure 9b).
>
> R: *In Figure 10, FFHQ, why does the model trained from scratch diverge? I would have expected StyleGAN2 to perform very well on FFHQ since this is the dataset it was designed for.*
>
> We argue that our runs are quite similar to the runs reported in [1] (Figure 1a) for the original StyleGAN2. While StyleGAN2 achieves remarkable quality on FFHQ, it still tends to get worse after a certain training step.

---

> > ### Comment · Reviewer_BE8y · 2021-11-25
> > **Response to Rebuttal**
> >
> > Thanks for running these additional experiments and adding them to the paper! I think these are useful datapoints to have for anyone who might want to explore GAN pretraining in the future.
> >
> > Regarding the divergence with FFHQ, I see now that the training plot does match the one in [1] when considering the 70k images with no ADA setting, thank you for pointing this out. What was the motivation for not using ADA when the dataset size was more than 50k? Also, why only use half of the FFHQ dataset and not the full 140k image set?

---

> > > ### Author Response · Authors · 2021-11-26
> > > **Setup Clarification**
> > >
> > > *What was the motivation for not using ADA when the dataset size was more than 50k?*
> > >
> > > For large datasets, using ADA can deteriorate the performance. For instance, see Figure 7c in [1] (LSUN-Cats line). Therefore, we employ ADA for small datasets only.
> > >
> > > *Also, why only use half of the FFHQ dataset and not the full 140k images set?*
> > >
> > > The original FFHQ dataset contains 70k images and 140k in [1] refers to the dataset populated with horyzontal flips. We did not use horizontal flips for FFHQ to have the same setup as for other datasets, which typically do not use flips.
> > >
> > > [1] Training Generative Adversarial Networks with Limited Data, Terro Karras et. al.

---

> > > > ### Comment · Reviewer_BE8y · 2021-11-26
> > > > **Response**
> > > >
> > > > Thank you for the clarification!

---

### Decision · Program_Chairs · 2022-01-20

**Decision:**

Accept (Poster)

**Comment:**

This paper empirically studies when, why, and which pretrained GANs are useful.
All the reviewers are positive about this work, that they all consider very valuable for practitioners and the community.
First building intuition through toy examples, authors conduct a large-scale study of transfer learning in GANs (with the stylegan2). They propose a way to understand the relevance of a pre-trained generator and discriminator, as well as heuristics to select good initialization.
Overall, this paper makes a solid contribution that should to be accepted.